# Assessing the Impact of Biodiversity (Species Evenness) on the Trophic Position of an Invasive Species (Apple Snails) in Native and Non-Native Habitats Using Stable Isotopes

**Kevin E. Scriber II** [1,*] **, Christine A. M. France** [2] **and Fatimah L. C. Jackson** [3]

1   Department of Environmental Science, University of Arizona, Tucson, AZ 85721, USA
2   Smithsonian Museum Conservation Institute, 4210 Silver Hill Rd., Suitland, MD 20746, USA; francec@si.edu
3   Department of Biology, Howard University, Ernest Everett Just Hall 415 College St., NW,
    Washington, DC 20059, USA; fatimah.jackson@howard.edu
*   Correspondence: kevescriber@arizona.edu; Tel.: +1-205-460-2349

**Abstract:** Invasive apple snails negatively impact non-native habitats and human well-being. Here, the trophic position of *Pomacea canaliculata* in native habitats (Maldonado, Uruguay) and non-native habitats (Hangzhou, China and Hawaii, USA) are compared. Detritus samples and tissue samples from apple snails were collected in all sites. Trophic levels were calculated as the difference between the mean $\delta^{15}$N values of detritus samples and corresponding apple snail tissue samples, divided by the mean $\delta^{15}$N fractionation for nitrogen per trophic level in freshwater habitats. The mean $\delta^{15}$N values of detritus in sites served as a baseline (i.e., zero trophic level), allowing direct comparisons. Linear regression analysis established a correlation between species evenness and apple snail trophic level ($R^2$ = 0.8602) in line with a Pearson's product-moment correlation value (−0.83) and 95% confidence interval (−0.87, −0.77). Normal quartile plots indicated two normally distributed subsets of apple snail trophic-level data: (1) a biodiverse subset containing the Uruguayan and Chinese lake sites and (2) the homogenized Hawaiian and Chinese creek sites. A precipice value for species evenness (separating biodiversity from homogenization), between (3.7) and (2.4), once descended to or surpassed separates statistically distinct, normal distributions of invasive apple snail trophic-level data from diverse versus homogenized habitats.

**Keywords:** apple snail; biodiversity; invasion; isotopes; *Pomacea*

## 1. Introduction

Biodiversity [1] encompasses the concepts of species diversity [2], taxonomic diversity [3], and community diversity [4], the metrics by which it is often assessed. Trophic ecology provides a framework to investigate biodiversity's impact on ecosystem function [5]. Considering pertinent studies of biodiversity and ecosystem function [6–9] in the Anthropocene [10,11], trophic ecology is a useful approach for preserving or improving biodiversity and ecosystem function now and in the future [12,13]. This study employed the diversity index of species evenness [14–17], which defines the relative abundance of species, to measure the trophic position of the invasive apple snail *Pomacea canaliculata* in native and non-native habitats. The contributory relationships between biodiversity loss and altered ecosystem processes [18], function, and stability [19–21] became evident.

Introduced species are transported outside of their native range by humans, and once established in non-native habitats, are likely to become invasive pests that result in economic or environmental harm [22], adversely impacting agriculture and/or natural ecosystems [22]. Invasive species often lead to homogenization, a process which alters habitats by anthropogenic forces. Through homogenization, native species are extirpated and replaced by non-native species [23–26]. This process, homogenization, generally increases the similarity of species and reduces biotic resistance to ecosystem change [27]. Invasive species directly and

indirectly impact trophic interactions in non-native communities [28] via the establishment of new trophic interactions or the alteration or eradication of previously existing ones, thereby reducing the overall complexity or stability of non-native habitats [29].

The Ampullariidae are commonly called apple snails [30], including a well-known invasive species, *Pomacea canaliculata*. This species is considered a voracious omnivore [31–35]. Surprisingly, where *P. canaliculata* are naturally distributed [30], they represent major components of freshwater biodiversity, serve as ecosystem engineers, and contribute to nutrient turnover, while serving as prey for other species [35–41]. Generally, invasive apple snails lack adverse agricultural or ecological impacts in native habitats [42] (p. 215). Despite the importance of the apple snail in native habitats, *Pomacea canaliculata* is counted amongst 100 of the world's worst invasive species [43] and is a major agricultural pest of aquatic staple crops such as rice, taro, and watercress [44–46]. *P. canaliculata* also shifts non-native freshwater habitats to alternate eutrophic states [47], adversely impacting production and environmental heterogeneity, as bottom-up controls act altering community structure, thereby diminishing biodiversity in these newly homogenized freshwater habitats [31,48]. Why apple snails differentially impact native versus non-native habitats is not well understood. Comparisons of the trophic position of *P. canaliculata* between native and non-native habitats could reveal a change in trophic position as a key factor influencing their adverse impacts as invasive species.

Stable isotope analysis (SIA) is widely implemented in ecological studies over a range of scales of biological organization to investigate ecosystem function in terms of animal migration patterns [49,50], resource availability [51,52], and even the dynamics of carbon, nitrogen, and water in ecosystems [53]. Stable isotope abundances for $^{15}N/^{14}N$ (heavy/light isotopes), when compared amongst species in an ecosystem, the ratio can be used to define their trophic position, indicating the trophic level on which species feed. This study assesses whether the relationship between species evenness (a measure of homogenization within habitats) and the calculated trophic level of invasive apple snails is linear. If so, species evenness may be used as an independent variable to predict trophic position, and additionally, stable isotopes can be used to indirectly infer the diet [54] of invasive species (using the invasive apple snail as a model organism).

## 2. Materials and Methods

### 2.1. Field Collection Sites

Apple snails were collected at five collection sites in three countries.

A.  Maldonado, Uruguay

(1) A pristine site at Lake Sauce and (2) Lake Dario, an anthropogenically disturbed site in the native range of *P. canaliculata*, where samples were subsequently processed at the CURE (Central University of the Republic) in December 2014.

B.  China

(3) A lake site and (4) creek site within XiXi National Park in Hangzhou, Zhejiang, China invaded by *P. canaliculata*, where samples were subsequently processed at the Zhejiang Provincial Key Laboratory of Biometrology and Inspection & Quarantine in July 2017.

C.  Oahu, Hawaii, USA

(5) Kawainui Marsh, a culturally important site to the Hawaiian people, invaded by *P. canaliculata*, where samples were subsequently processed at the Bernice Pauahi Bishop Museum, Oahu, Hawaii in November of 2018.

Specific coordinates and details on each collection site are provided in the Supplementary Materials Section S.1.

### 2.2. Apple Snails, Other Sympatric Animal Species, and Detritus

*Pomacea canaliculata* and other sympatric animal species (species that live in the same geographic location) were collected by hand using small aquatic nets or aerial nets. In the native habitat (Maldonado, Uruguay), field collections of all species except fish occurred over 3 days with 7 individuals working for 4 h per day. Fish were collected once at each native site using seine nets by 7 individuals for 1 h. In the Chinese collection sites (Hangzhou, China), field collections were made at both sites over 3 days with 4 individuals working for 6 h per day. Field work in the Kawainui marsh was conducted over 1 day by 1 individual over 8 h.

Once collected, all specimens were inventoried, frozen, and thawed. Genetic samples from all *Pomacea canaliculata* and other metazoan animal species were stored in 95% ethanol, while the remaining material to be used for stable isotope analysis was rinsed with deionized water, weighed, dried at 60 °C for a minimum of 24 h, subsequently re-weighed, the dried mass recorded, and then ground into a homogeneous powder for stable isotope analysis.

Detritus was sampled using 42-cm-tall plastic cylinders with a diameter of 15 cm. These cylinders were pressed into the substrate by hand. The contents of the cylinder were pulled from the water and deposited into labeled plastic bags and later filtered by hand, using a three-piece Soil Sieve Set (1″, 0.2″, and 0.25″ mesh diameter) and running deionized water. Detritus was then dried at 60 °C for 24 h, weighed, and subsequently ground into a homogeneous powder for stable isotope analysis. *P. canaliculata* and other macroinvertebrate stable isotopes and corresponding genetic samples, as well as detritus samples, were stored at −20 °C and later transported to Howard University for long-term storage at −80 °C.

### 2.3. Species Barcoding

Total genomic DNA was extracted from all apple snails (*Pomacea* spp.), arthropods, other miscellaneous soft-bodied invertebrates, and fish samples using the DNeasy Blood and Tissue Kit (Qiagen, Hilden, Germany). The QIAGEN Supplementary Protocol "Purification of total DNA from insects using the Dneasy Blood & Tissue Kit" was used with a maximum mass of 50 mg for extraction.

### 2.4. Polymerase Chain Reaction (PCR) Amplification

Genomic DNA samples from *P. canaliculata* and other metazoans were utilized in polymerase chain reactions with the Bioline Company (Memphis, TN, USA) molecular reagents. Each PCR reaction had a total volume of 25 ML, composed of diH20 (9.25 ML), Bioline 5X Buffer (No Mg, 5 ML), MgCl$_2$ (2.5 Mm, 1.25 ML), dNTPs (0.2 Mm, 4 ML), a forward primer (0.15 Mm, 1 ML), a reverse primer (0.15 Mm, 1 ML), BSA (0.4 µg/ML, 1 ML), DMSO ([0.5%], 0.125 ML), Mango Taq (5 U/ML, 0.2 ML), and template DNA (2 ML).

A Gene Touch PCR thermocycler with two-bay 96-well heat block was used to run all PCR programs. The mitochondrial cytochrome *c* oxidase subunit was targeted to barcode all taxa [55] and compare sequences obtained against available barcode databases using Geneious Prime 2019. A number of primers were used to amplify the variety of animal species collected. These Cytochrome Oxidase Subunit I primers are listed here (with the **5′** = Five Prime versus **3′** = Three Prime ends of these nucleic acids labeled as such):

1. **LCO1490 Universal Arthropod Primer [56]**
2. **HCO2198 Universal Arthropod Primer [56]**
3. **COXAR Caenogastropoda Primer [57]**
4. **FFD2 Universal Fish Primers [58]**
5. **FR1D Universal Fish Primers [58]**

All paired primers and the associated thermocycle for the species they are meant to amplify are displayed below:

Arthropod Species

**Primers: LCO1490 and HCO2198 [56]**

| 5 Min | 1 Min | 1 Min | 30 Sec | 20 Sec | 25 Sec | 10 Min | |
|-------|-------|-------|--------|--------|--------|--------|-----------|
| 95 °C | 45 °C | 72 °C | 95 °C | 48 °C | 72 °C | 4 °C | 50 Cycles |

**Apple Snails (*Pomacea* spp.) and other molluscs**
**Primers: LCO1490 and COXAR [57]**

| 5 Min | 1 Min | 1 Min | 30 Sec | 30 Sec | 45 Sec | 10 Min | |
|-------|-------|-------|--------|--------|--------|--------|-----------|
| 95 °C | 45 °C | 72 °C | 95 °C | 48 °C | 72 °C | 4 °C | 75 Cycles |

Fish Species
**Primers: FFD2 and FR1D [58]**

| 5 Min | 1 Min | 1 Min | 30 Sec | 30 Sec | 45 Sec | 10 Min | |
|-------|-------|-------|--------|--------|--------|--------|-----------|
| 95 °C | 45 °C | 72 °C | 95 °C | 48 °C | 72 °C | 4 °C | 60 Cycles |

Successfully amplified PCR products were shipped to Eurofins Genomics (Louisville, KY, USA) for sequencing. The resultant sequences were edited and identified to species, or the most concise taxonomic level possible, using the Geneious Prime (2019 version) (New Zealand) and the NCBI (National Center for Biotechnology Information (Rockville, MA, USA)) database. An animal collection catalogue, as well as the corresponding means of for the identification of animals collected and corresponding sequence data, where applicable, is available in Supplementary Materials Sections S.3 and S.4, respectively.

### 2.5. Stable Isotope Analysis ($^{13}C$ and $^{15}N$)

Stable isotope analysis was performed at the Smithsonian Institution's MCI Stable Isotope Mass Spectrometry Laboratory (Suitland, MD, USA). Samples were weighed into tin cups and run on a Thermo Delta V Advantage mass spectrometer in continuous flow mode, coupled to a Costech 4010 Elemental Analyzer (EA) or an Elementar Isotope Cube via a Thermo Conflo IV. All calculations of raw isotope values were performed with Isodat 3.0 software.

Raw isotope values were calibrated using a 2-point correction against a Costech Acetanilide and a urea (Urea-UIN3) [59], which were both calibrated to USGS40 (L-glutamic acid) and USGS41 (L-glutamic acid). All data are reported using standard delta notation: $\delta^{15}N = [(^{15}N/^{14}N_{sample} - ^{15}N/^{14}N_{standard})/^{15}N/^{14}N_{standard}] \times 1000$, where the standard is atmospheric air and units are per mil (‰). The $\delta^{13}C$ values are calculated similarly, where the ratio of interest is $^{13}C/^{12}C$ and the standard is V-PDB. Reproducibility was $\leq 0.2‰$ (1σ) based on repeated measures of reference materials and select samples. Error associated with the resulting stable isotope sample data points was reported as $\pm 0.2‰$.

### 2.6. Stable Isotope Data Analysis and Trophic-Level Determination and Evenness

The resulting $\delta^{13}C$ and $\delta^{15}N$ values reported for all stable isotope samples collected from native and non-native habitats were compared using an ANOVA (analysis of variance, using the R statistical program). The trophic level of *P. canaliculata* in the respective habitats was calculated by subtracting the mean $\delta^{15}N$ values of detritus samples from corresponding mean $\delta^{15}N$ values for *P. canaliculata* collected within each habitat, and subsequently dividing the differences, for each site, by an expected $\delta^{15}N$ fractionation per trophic level 2.98‰ $\pm$ 0.11 SD; see [60]. The $\delta^{15}N$ values of detritus in each site provided a baseline reference (i.e., a zero trophic level), thereby allowing the trophic

level of *P. canaliculata* to be compared between these distinct habitats. This procedure allowed the trophic position of *P. canaliculata* to be compared directly between (native and non-native) habitats.

The biodiversity metric of species evenness was calculated as:

**Evenness =
1/(# of the most abundant species collected/ total # of organisms collected)**

Evenness values from each collection site were compared to the trophic level of *P. canaliculata* using linear regression to ascertain whether species evenness can explain variation in the trophic position of *P. canaliculata* between the five collection sites. Specifics on the enumeration of animal species catalogued in each collection site and subsequently calculated species evenness values, as shown above, are provided in the Supplementary Materials Section S.3.

## 3. Results

### 3.1. Linear Regression of Species Evenness versus Trophic Level for All Collection Sites

Stable isotope analysis showed that mean $\delta^{15}N$ values were highest in Kawainui Marsh and decreased in order between the Chinese Creek Site, Lake Sauce, the Chinese Lake site, and finally Lake Dario. The standard deviation of $\delta^{15}N$ values was also greatest in the Chinese creek site (see summarized data in Table 1, complete data set in the Supplementary Materials Section S.2). As this study relies most heavily on $\delta^{15}N$ values for subsequent analyses, the $\delta^{13}C$ values are presented only in the Supplementary Materials Section S.2 alongside their corresponding $\delta^{15}N$ values for all individual apple snails collected in respective sites. Species evenness was greatest in Lake Dario and decreased in order from Lake Sauce to the Chinese lake site, Kawainui Marsh, and the Chinese creek site (see Table 1). The calculated trophic level of *Pomacea canaliculata* was lowest in Lake Dario. Corresponding trophic-level data from the remaining sites increased from Lake Sauce to the Chinese lake site, to Kawainui marsh, and were greatest in the Chinese creek site, where the variation in the trophic level of *Pomacea canaliculata* was also greatest (see Table 1).

**Table 1.** The mean $\delta^{15}N$ (‰, air) and stdv. (standard deviation) for *P. canaliculata* and detritus in collection sites, the species evenness in collection sites, and the trophic level and stdv. of *P. canaliculata* in collection sites.

| SIA Sample | Site | $n$ | $\delta^{15}N$ (Mean) (‰, Air) | $\delta^{15}N$ (stdv) (‰, Air) | Species Evenness | *P. canaliculata* Trophic Level | *P. canaliculata* Trophic Level (stdv) |
|---|---|---|---|---|---|---|---|
| *P. canaliculata* | Lake Dario | 64 | 2.0 | 0.9 | 4.9 | 0.3 | 0.3 |
| Detritus | Lake Dario | 5 | 1.3 | 0.8 | 4.9 | | |
| *P. canaliculata* | Lake Sauce | 50 | 4.9 | 0.9 | 4.6 | 0.7 | 0.3 |
| Detritus | Lake Sauce | 5 | 2.8 | 0.8 | 4.6 | | |
| *P. canaliculata* | Chinese Lake | 50 | 4.0 | 1.0 | 3.7 | 0.4 | 0.3 |
| Detritus | Chinese Lake | 5 | 2.9 | 1.0 | 3.7 | | |
| *P. canaliculata* | Kawainui | 17 | 8.1 | 1.1 | 2.4 | 1.6 | 0.4 |
| Detritus | Kawainui | 5 | 3.4 | 1.4 | 2.4 | | |
| *P. canaliculata* | Chinese Creek | 18 | 6.2 | 2.1 | 2.2 | 2.1 | 0.7 |
| Detritus | Chinese Creek | 5 | −0.1 | 3.3 | 2.2 | | |

Species evenness from the collection sites was correlated with the calculated trophic level of *P. canaliculata* therein. Linear regression showed species evenness explained 68.2% of trophic-level variation for *P. canaliculata* (Figure 1). A subsequently performed linear model (relating (1) species evenness and (2) calculated trophic-level data for *P. canaliculata*) produced similar $R^2$ (0.68) and adjusted $R^2$ values (0.68), while a highly significant ($p < 0.0001$) correlation test also produced a Pearson's product-moment value of ($-0.83$, see Table 2).

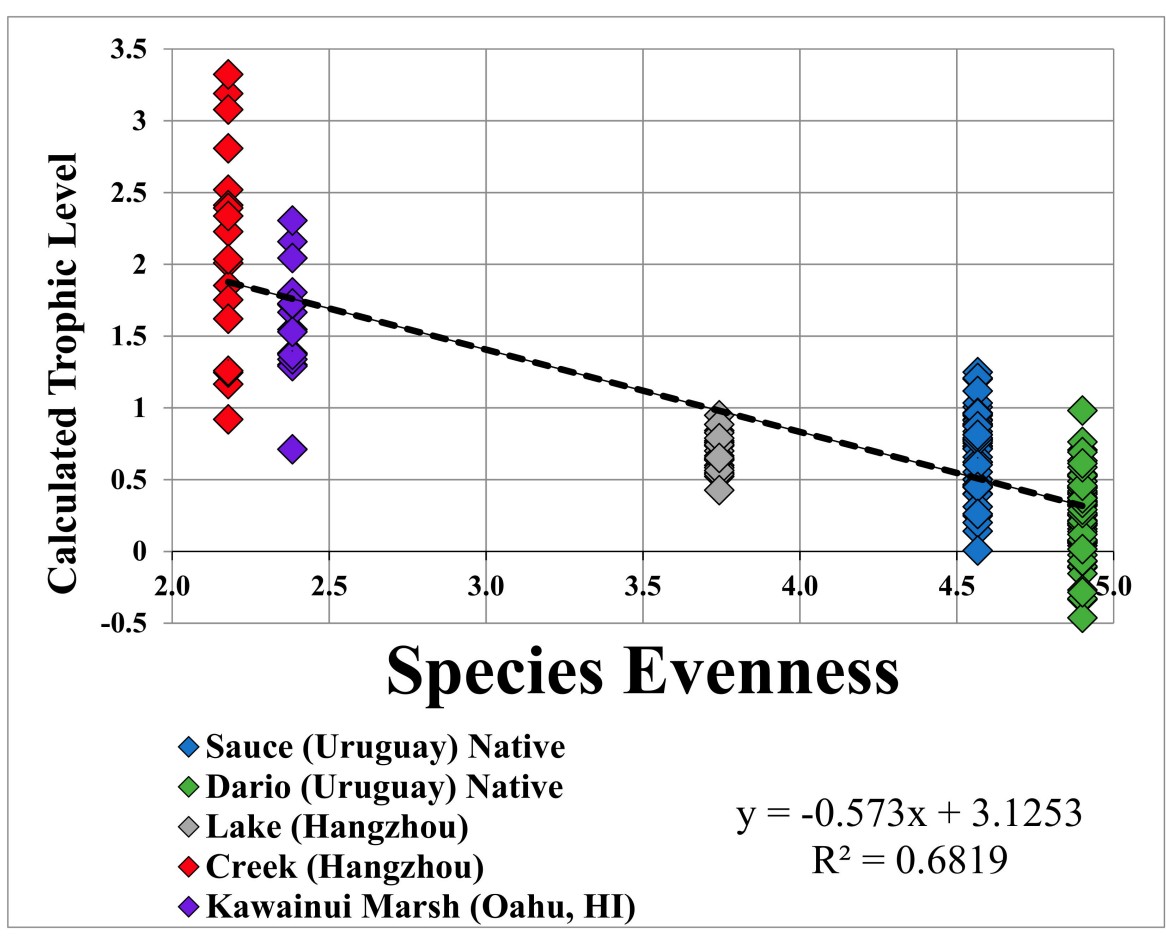

**Figure 1.** The estimated trophic level of *P. canaliculata* as a function of species evenness for each collection site. The two native habitats (Lake Sauce and Dario) are labeled.

**Table 2.** Results of (1) a linear regression model and (2) the Pearson's product-moment correlation for: (a) species evenness and (b) *P. canaliculata* trophic levels in collection sites.

| Statistical Analysis | Results | | |
|---|---|---|---|
| Linear regression model | $p$-value < 0.001 | $R^2$ = 0.6821 | Adjusted $R^2$ = 0.8602 |
| Pearson's product-moment correlation | $p$-value < 0.001 | 95% confidence interval for correlation coefficient ($-0.87$, $-0.77$) | Correlation coefficient $-0.83$ |

### 3.2. Interquartile Plots of Calculated Trophic Levels for P. canaliculata

Calculated trophic-level estimates were checked for normality. Using a normal quantiles plot across all collection sites, the majority of these trophic-level estimates were found to be normally distributed (Figure 2). Those trophic-level estimates that did not fit a normal distribution were from two sites—those with the lowest values of species evenness—Kawainui Marsh and the Chinese creek site.

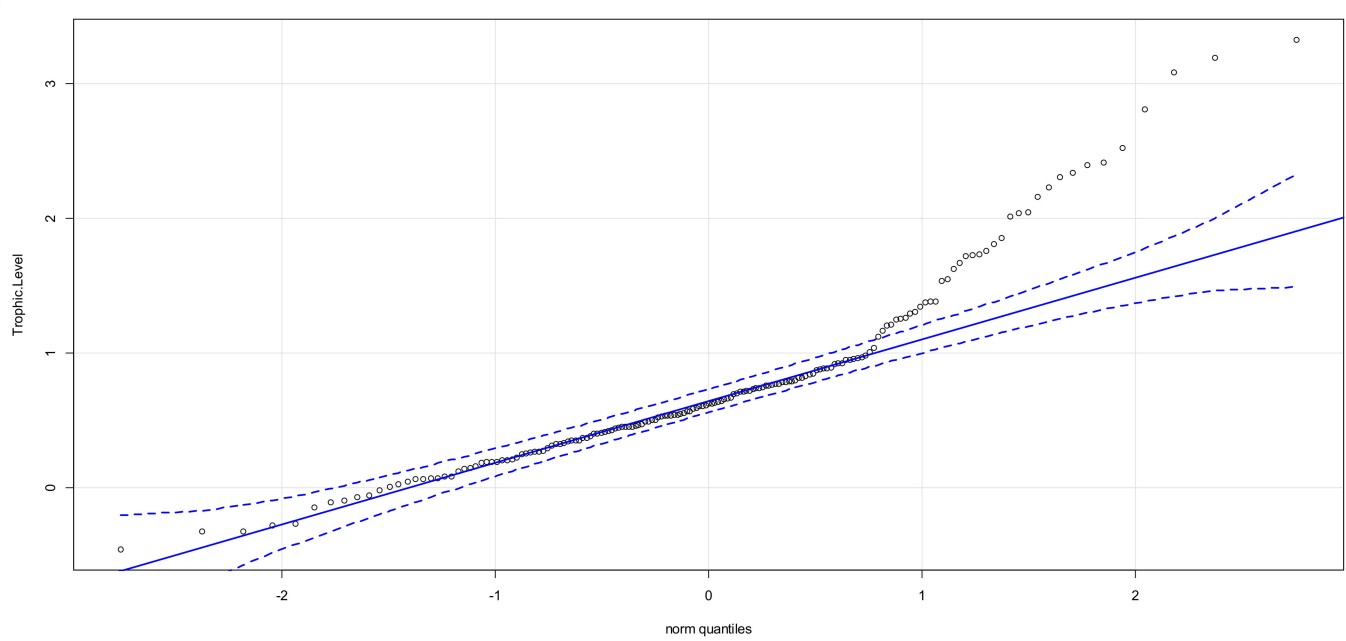

**Figure 2.** Normal quantiles plot testing the normality of all calculated trophic-level estimates for *P. canaliculata* from all collection sites, with (blue dotted lines) 95% confidence intervals depicting the limits of a normally distributed data set.

Figure 2 illustrates that the trophic-level data from all five collection sites, when taken together, diverge from normality and that a subset of those data is statistically distinct. In Figure 2, the *x*-axis is a normal quartile scale, and the *y*-axis is a compendium for the trophic-level data from all the collection sites; the dotted blue lines represent the 95% confidence intervals for a normal distribution of data. It therefore illustrates the desired test for data normality across all collection sites.

As a whole, the stable isotope data from all collection sites explained 68% of the variation observed in trophic-level variation (Figure 1). These trophic-level estimates, taken together, did not fit a normal distribution (see Figure 2) and were utilized in two subsequent normal quantile plots (Figures 3 and 4) that separated trophic-level estimates, which initially did not diverge from normality from those that did. The first subset of trophic-level estimates was collected in Lake Sauce, Lake Dario, and the Chinese lake site (see Figure 3); the second subset of trophic-level estimates was collected in the Chinese creek site and in Kawainui Marsh (see Figure 4). Separating samples into two normal quantiles plot testing the normality of all calculated trophic-level estimates demonstrated the corresponding trophic-level estimates comprised two sets of normally distributed data, one where the collections sites' species evenness was $\geq 3.7$ (Figure 3) and another where the species evenness was $\leq 2.4$ (Figure 4), respectively.

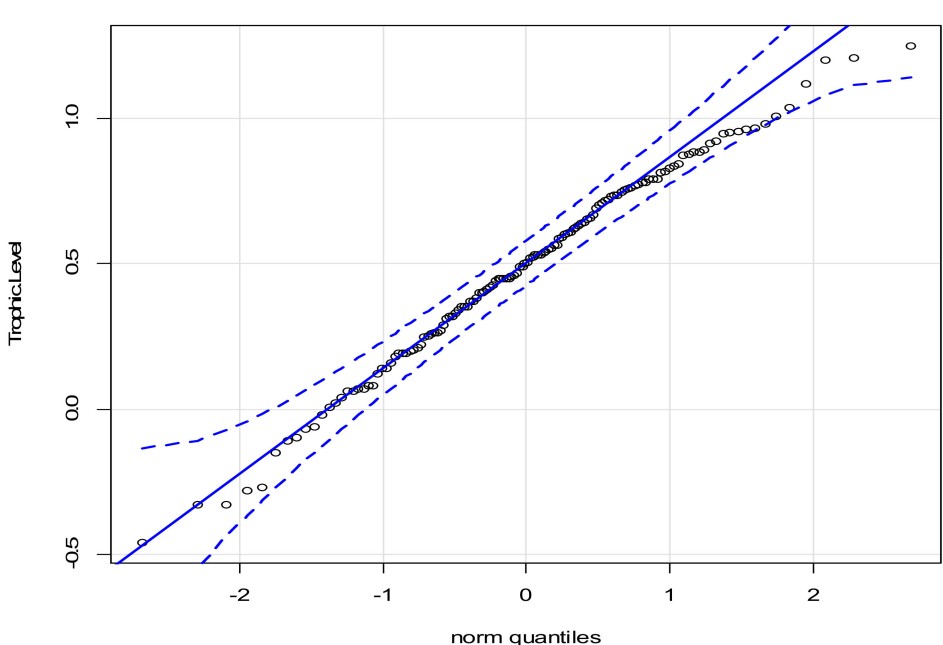

**Figure 3.** Normal quantiles plot testing the normality of all calculated trophic-level estimates for *P. canaliculata* collected in Lake Sauce, Lake Dario, and the Chinese Lake site (here species evenness was $\geq$3.7), with (blue dotted lines) 95% confidence intervals depicting the limits of a normally distributed data set.

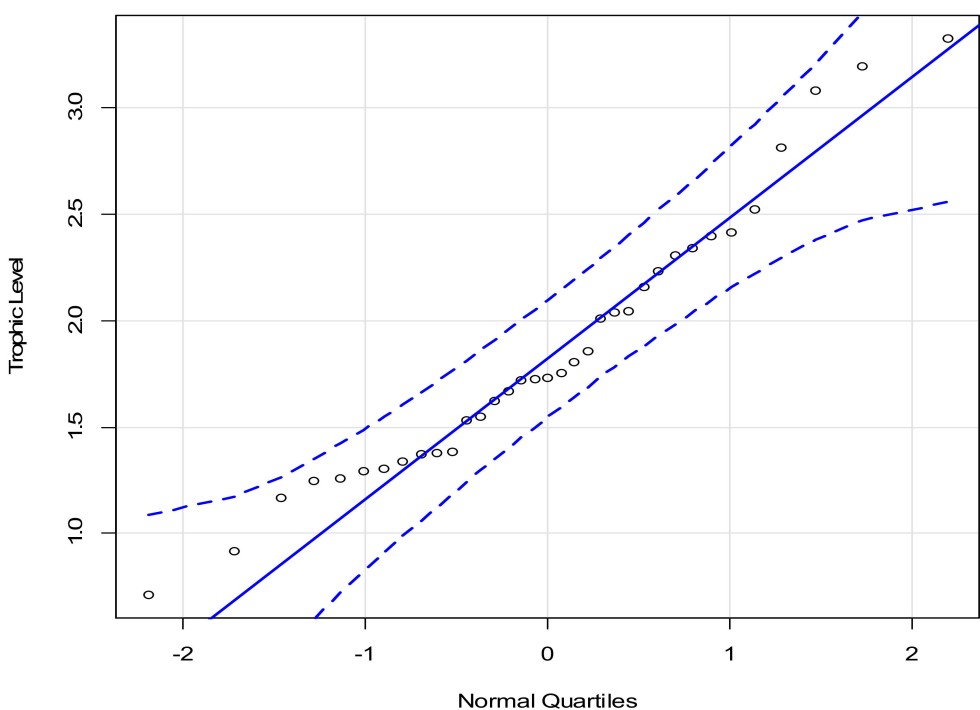

**Figure 4.** Normal quantiles plot testing the normality of all calculated trophic-level estimates for *P. canaliculata* from Kawainui Marsh and the Chinese creek site (here species evenness was $\leq$2.4), with (blue dotted lines) 95% confidence intervals depicting the limits of a normally distributed data set.

The value of species evenness, our metric of homogenization (biodiversity loss) within all five collection sites, decreases from Lake Dario (4.9) to Lake Sauce (4.6), to the Chinese Lake site (3.7), in the first subset of normally distributed data (Table 1 and Figure 3). Species evenness

continues to decline in the second subset of normally distributed data, with Kawainui Marsh (2.4), reaching a minimum value in the Chinese creek site (2.2; Table 1 and Figure 4).

Figure 5 illustrates how the calculated trophic level of these invasive apple snails departs more from normality while moving from native to non-native, and increasingly homogenized, environments. This eventually culminates in the loss of normality in the Kawanui marsh, which was the most homogenous collection site of all (as indicated by the outliers beyond the 95% confidence interval).

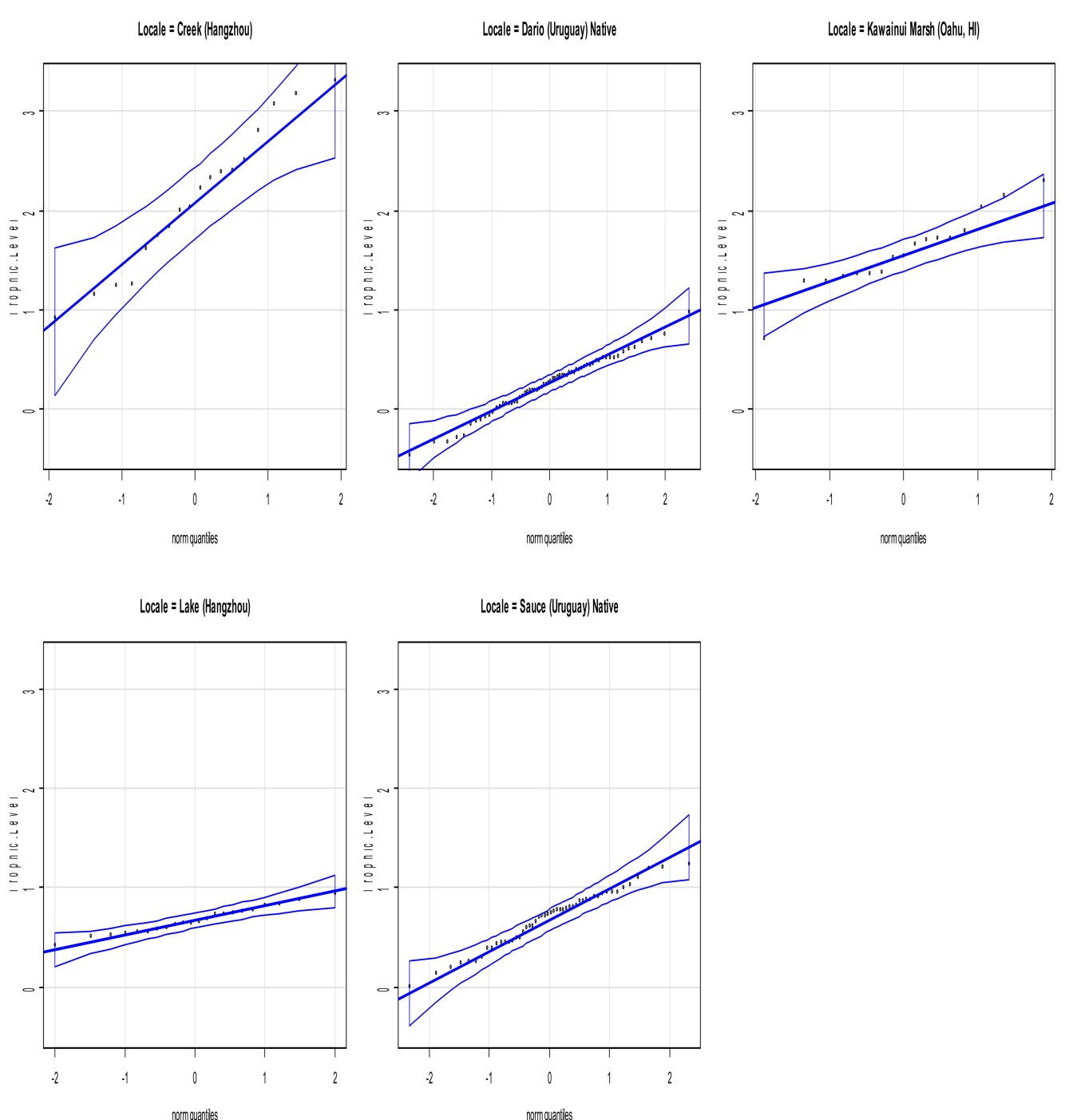

**Figure 5.** Normal quantiles plot testing the normality of all calculated trophic-level data for *Pomacea canaliculata* from each collection site individually, with (blue polygons) 95% confidence intervals depicting the limits of normally distributed data within each site.

## 4. Discussion

Our results (as illustrated by Figure 1 and Table 1) indicate that species evenness in native and non-native habitats is a good predictor of the average calculated trophic level of *P. canaliculata*, explaining more than 68% of the variation observed. These results indicate a strong linear relationship between species evenness (as an independent variable) and the calculated trophic level of this invasive freshwater species. What is likely more telling is the realization that these data from five unique collection sites can be placed into two normally distributed subsets (as shown by normal quartile plots in Figures 2–4).

The trophic levels depicted in Figure 1 are estimates based on the detritus stable isotope data. Lake Dario was anthropogenically disturbed by the removal of macrophytes, leading to the alteration of production and the accrual of detrital material in the lake. This may explain why a subset of apple snails was nitrogen-poor therein, resulting in a calculated trophic level below the baseline (or zero trophic level), as seen in Figure 1.

In the cases of these specific snails, they had less $^{15}$N than the detritus samples average within that specific site (as listed in Table 1), hence the negative value for their respective trophic level. This anomaly was only evident in Dario and only in a subset of snails. The high level of species diversity (high species evenness value) and the diminished production, due to anthropogenic influences within Lake Dario, together may account for this apparition in the resultant trophic levels of a subset of snails in Lake Dario.

Figures 3 and 4, respectively, test two distinct subsets of three and two collection sites for normality independently. Again, Figures 3 and 4 display 95% confidence intervals, which illustrate that these two distinct subsets of data do not diverge from normality. While when graphed individually, only one collection site, Lake Sauce (the least biodiverse site of all), diverged from normality using the same analysis. This may indicate a change in the relationship between the independent variable used in the linear regression (Figure 1, species evenness) and the calculated trophic level. An explanation for this may be that invasive apple snails in less diverse habitats may occupy open niche space formally occupied by organisms extirpated from these respective habitats as biodiversity diminished.

One explanation for this phenomenon may be that there is a tipping point, or threshold value, for species evenness where the level of homogenization outweighs the remaining biodiversity in habitats (whether native or non-native). In between the corresponding values for species evenness observed in the Chinese Lake site (3.7) and Kawainui marsh (2.4), this threshold exists for invasive apple snails.

Once met (e.g., as species evenness is reduced to and/or below this tipping point) the trophic level of *Pomacea canaliculata* therein would diverge from normality with corresponding data from: (1) the native habitat of invasive *Pomacea canaliculata* (Lake Sauce and Lake Dario), where they have no apparent negative ecological impacts, or (2) the less biodiverse, but statistically indistinct (see Figure 3), Chinese Lake site.

These five collection sites are separated into two categories: (1) the more biodiverse and less homogenized subset of statistically indistinct collection sites (Lake Sauce, Lake Dario, and the Chinese Lake site; see Figure 3); and (2) the less biodiverse and more homogenized collection sites (Kawainui Marsh and the Chinese creek site; see Figure 4). These data, from all five collection sites, suggest a useful approach for predicting the trophic level occupied by *Pomacea canaliculata* in native and non-native habitats, and possibly where the negative ecological impacts they have in less diverse, more homogenized habitats, will occur over broad geographical distances (see Figure 1 and Table 2).

These analyses support the following prediction: where species evenness values, and therefore biodiversity, are low (e.g., in the Chinese Creek site and the Kawainui Marsh site) *Pomacea canaliculata* occupy trophic levels higher than the corresponding snails collected in sites where species evenness values, and therefore biodiversity, are high (e.g., in the Lake Sauce, Lake Dario, and the Chinese Lake site). Hence, the calculated trophic level of this invasive species, *Pomacea canaliculata*, is inversely related to species evenness in native versus non-native habitats where it is found; second, the occupancy of higher trophic levels may be directly tied to the known adverse ecological impacts of invasive apple snails.

Changes in apple snail diets from more generalist specialist feeding behaviors have been reported previously [54], which fall in line with the findings of this study. What is needed is:

1.   An investigation of the possible relationship between the nutritional value of the plant species that comprise the apple snail diets (as previously determined using SIAR mixing models by Scriber, France, and Jackson [54])
2.   As well as an investigation into the differences in the variability ($s^2$) of the trophic level occupied by these aquatic invasives to test the hypothesis that they ecological niche they occupy in aquatic habitats:
     a.   Not only changes in the calculated trophic level of this aquatic biological invader between habitats, shown to exist by this study.
     b.   But also the constriction or expansion of their ecological niche space quantified as significant differences amongst the calculated trophic-level sample variance $s^2$ (a point estimate of $\sigma^2$) between habitats.

We note that sample variance is compared most appropriately via a Levene's test, and possibly subsequent pair-wise comparisons of sample variance ($s^2$) using a Tukey test, if at least one population variance ($\sigma^2$) is found to be significantly different in the initial analysis.

In concert with the findings of this study, further studies of this type would provide valuable insights as to how and why the calculated trophic level of invasive species changes between native versus non-native habitats, alongside diet. It will also further our understanding of the role that species evenness (homogenization), and possibly open niche space resulting from homogenization, has on changes in diet and trophic position between these respective habitats.

The noticeable increase in the calculated trophic level of *Pomacea canaliculata* in less biodiverse or homogenized habitats, which here comprised two non-native habitats (e.g., Kawainui Marsh and the Chinese creek sites), provides a mechanism by which *P. canaliculata* responds to reduced interspecific competition (and/or increased intraspecific competition during novel biological invasions) for essential resources (e.g., food and space).

Reduced interspecific competition in less biodiverse (more homogenized) or less productive habitats consequentially may shift the trophic position of *Pomacea canaliculata*. This trophic shift may cause the well-known adverse ecological impacts associated with *P. canaliculata* that extirpate both aquatic and riparian plants in invaded habitats and facilitate trophic cascades via the bottom-up control of community structure [1,7]. This process ultimately contributes to further loss of biodiversity and increasing homogenization therein.

## 5. Conclusions

This study provides data that support the proposed mechanism by which the adverse ecological impacts of *Pomacea canaliculata* in non-native habitats are explained. These data also provide a means of predicting the susceptibility of non-native habitats to the negative ecological impacts of this aquatic invasive species. As biodiversity declines within non-native habitats, the arrival of invasive species readily fills open niche space within these habitats.

What may be more significant is that the data illustrate a clear relationship between reduced biodiversity in habitats and the prevalence of adverse ecological impacts from invasive apple snails. These adverse ecological impacts differ starkly from the impacts of invasive apple snails in pristine and/or anthropogenically disturbed portions of the native range, as well as in more biodiverse and less homogenized non-native habitats (e.g., the Chinese Lake site), where any deleterious effects of *Pomacea canaliculata* are absent or less obvious.

The fact that *Pomacea canaliculata* can have variable ecological impacts in native versus non-native habitats suggests that this proposed mechanism of trophic shifts may explain apparent changes in invasive apple snail feeding behavior and/or diet between habitats. It appears that the trophic shifting, as suggested by the data presented in this publication, demonstrates that as this invasive species moves from (1) biodiverse native habitats, to (2) anthropogenically disturbed native habitats, and (3) finally to more and more homoge-

nized (and therefore less biodiverse) non-native habitats, may correspond to a shift in their ecological niche, diet, and inherent (natural versus unnatural) ecological role.

We suggest this is a plausible explanation as to how invasive apple snails transition from being primarily opportunistic generalists, in highly productive and relatively biodiverse native habitats within the Pantanal wetland of Brazil, to macrophyte and/or riparian plant specialists in more homogenized, less biodiverse and by extension less interspecifically competitive, non-native habitats. Stable isotope studies could be used preemptively, at the forefront of biological invasion, to determine habitats' susceptibility to biological invaders and the adverse ecological impacts predicted by the invasion front hypothesis [61]. Conversely, in habitats already suffering these ill ecological effects, stable isotope studies can provide insights into (1) the changes to habitats' trophic ecology which may facilitate these impacts and (2) determining the likelihood that habitats may be restored to proper function. The trophic level of *Pomacea canaliculata*, as well as other biological invaders, may be predictable based on differences in biodiversity (species evenness) and stable isotope data. This may prove to be a powerful tool in the conservation and/or restoration of habitats that invasive species endanger and/or inhabit, respectively. Future studies would benefit from evidence inferred from the stable isotope data collected within these five collection sites to examine the specific contribution of available food resources to the diet of biological invaders in general. In doing so, the observations from this study can be supported, or disproven, by indirectly inferring and defining the components and proportions that available food resources contribute to biological invader diets in native and/or non-native habitats along a continuum of species evenness (a measure of biodiversity or homogeneity).

**Supplementary Materials:** The following Supporting Information can be downloaded at: https://www.mdpi.com/article/10.3390/su15118560/s1, Section S.1: Study Collection Site Coordinates; Section S.2: Tables S1–S5 list of all individual detritus samples and apple snails (*Pomacea canaliculata*) from all five collection sites, and the corresponding $\delta^{15}$N and $\delta^{13}$C values and the calculated trophic level for all samples. Detritus samples served as a baseline for comparison between habitats; as such the net value of the trophic levels for these samples, within each collection site, is and should be zero; Section S.3: All animal species catalogued and identified via COI barcoding in all collection sites during this study; Section S.4: Identified species and respective COI consensus and reference sequences (NCBI).

**Author Contributions:** K.E.S.II conceived project design and designed methodologies, completed all fieldwork, prepared samples, analyzed samples, processed data, analyzed data, wrote, and edited all manuscript drafts. C.A.M.F. assisted with project design and methodologies, provided training, provided laboratory support, analyzed samples, assisted with data analysis, and edited all manuscript drafts. F.L.C.J. assisted with project design and methodologies, assisted with general project management, provided funding and laboratory space, assisted with data analysis, and edited all manuscript drafts. All authors have read and agreed to the published version of the manuscript.

**Funding:** Financial support in the form of a $1870.00 grant, specifically the 2018 Frederic Weiss Memorial Award, was provided by the Conchologist of America in support of the portion of this research conducted on Oahu, Hawaii, USA. Also the APC (Article Publication Charges) for this manuscript were covered in part, or entirely, by the Department of Environmental Science at The University of Arizona in Tucson, Arizona, USA.

**Institutional Review Board Statement:** Ethical review and approval were waived for this study because it solely involved the handling of invertebrate and plant species, as well as, planktonic organisms, by K.E.S.II. As such, the IACUC (Institutional Animal Care and Use Committee) at Howard University and the Howard University Graduate School exempted this project from IACUC review.

**Informed Consent Statement:** Not applicable, as human subjects were not used.

**Data Availability Statement:** Data are contained within the article and/or Supplementary Materials therein.

**Acknowledgments:** We acknowledge the help of Leslie Ries in the Department of Biology at Georgetown University for her support in completing my field work and the opportunity to share my research endeavors and receive critical and valuable feedback from her laboratory; undergraduate students from Howard University (Zahra Mansur and Brittany Galloway) for their help in completing my stable isotope analyses; the CURE Institute in Maldonado, Uruguay, Jiliang University in Hangzhou, China, Howard University's Department of Biology, The Bishop Museum on Oahu, Hawaii, and The Smithsonian Institution for support to complete fieldwork for this research. Last, we acknowledge funding from the Conchologist of America in support of this research in Hawaii.

**Conflicts of Interest:** There are no conflict of interest, financial or otherwise, that have influenced authors' objectivity towards the production and/or publication of this research.

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
