# Peer review of "Assessing the Impact of Biodiversity (Species Evenness) on the Trophic Position of an Invasive Species (Apple Snails) in Native and Non-Native Habitats Using Stable Isotopes"

_sustainability, doi:10.3390/su15118560_

Round 1

Reviewer 1 Report

This paper aims to assess the impact of biodiversity on the trophic position of an invasive species in native and non-native habitats using stable isotopes. Some comments for the authors to consider in revising the manuscript are as follows:

1) Abstract: Please add a sentence to conclude the findings at the end.

2) Materials & methods: To get a genuine samples in this kind of study is crucial. Any selection criteria for the sources or provenance of the samples? 

line 182, how much is the weight?

3) Table 1, would be good to justify the different number of samples from different locations.

4) Fig 1, the r2 is actually just moderate. How it affects the results in this study?

5) In discussion, would be good to include and associate it more to the diet and environment too, since the title mentioned about biodiversity.

6) General comments: Please check through the entire manuscript again for minor syntax errors.

Acceptable, but with some minor syntax errors.

Author Response

  • Abstract: Please add a sentence to conclude the findings at the end.

I believe the abstract explains the findings sufficiently. Please see quotation,

“Linear regression analysis established correlation between species evenness and apple snail trophic level (R2 = 0.8602); in line with a Pearson's product-moment correlation value (-0.83) and 95% confidence interval (-0.87, -0.77). Normal quartile plots indicated two normally distributed subsets of apple snail trophic level data: (1) a biodiverse subset containing the Uruguayan and Chinese lake sites and (2) the homogenized Hawaiian and Chinese creek sites. A precipice value for species evenness (separating biodiversity from homogenization), between (3.7) and (2.4), once descended to or surpassed separates statistically distinct, normal distributions of invasive apple snail trophic level data from diverse versus homogenized habitats.”

  • Materials & methods: To get a genuine samples in this kind of study is crucial. Any selection criteria for the sources or provenance of the samples?

I cannot understand the two sentences above, highlighted in yellow.

line 182, how much is the weight?

The mass of each sample was variable and dependent upon the mass of international standards using in the analyses, as well as if the samples were from plant versus animal species. The literature cited in this paper provides sufficient background for the replication of these methods, or their application to a similar study; provided the reader makes use of the referenced article on stable isotope analyses. I hope this explains why the specific masses utilized were not listed for every sample collected.

  • Table 1, would be good to justify the different number of samples from different locations.

There were different numbers of samples in each site as there were different numbers, and densities of snails therein. I do not think it is necessary to explain that; especially as the  number of samples collected in each site did not impact the power of the statistical analyses utilized. That is to say, the sample sizes all exceeded n=7.

  • Fig 1, the r2 is actually just moderate. How it affects the results in this study?

Though, the r2  value was 0.68, linear regression was subsequently utilized to bolster the strength  of the argument that diversity, as assessed by species evenness, and trophic level were correlated. So, as you see the linear regression and Pearson’s product moment and the calculated 95% confidence interval indeed bolstered that argument.

Linear regression analysis established correlation between species evenness and apple snail trophic level (R2 = 0.8602); in line with a Pearson's product-moment correlation value (-0.83) and 95% confidence interval (-0.87, -0.77).

            Nonetheless, the fact that the first figure showed a,” moderate”, relationship over vast geographical ranges, including the most isolated archipelago in the world, Uruguay in South America, and Hangzhou, Zhejiang, China, was a sufficient premise for the subsequent analysis. I hope this explains why 0.68 was a sufficient r2 value to prompt the continuation of this analysis.

5) In discussion, would be good to include and associate it more to the diet and environment too, since the title mentioned about biodiversity.

I do not see how discussing the diet of the diet of apple snails in these habitats at length would inform the reader as to the biodiversity therein, as the biodiversity was assessed by the metric of species evenness. Additionally, the prior publication in this journal resulting from this research explicitly defines the diet of the apple snail in each of these habitats respectively and is referenced herein. The onus here is on providing evidence of the existence of this relationship between biodiversity, as assessed by species evenness which is discussed at length, and the calculated trophic level of apple snail in these various collection sites.

6) General comments: Please check through the entire manuscript again for minor syntax errors.

I will review the paper for grammatical and/or syntax errors and I thank you for reading the manuscript.

Reviewer 2 Report

Though this short text raises important problems and provides original and potentially interesting results, I do not feel that the conclusions presented by the authors are substantiated by the design of their study. In other words, the reported relationship (“a clear relationship between reduced biodiversity in habitats and the prevalence of adverse ecological impacts from invasive apple snails”) seems not to be properly substantiated by the facts and, thus, remains unconvincing. I can agree with this claim: “where species evenness values, and therefore biodiversity, are low <…> Pomacea canaliculata occupy trophic levels higher than corresponding snails collected in sites where species evenness values, and therefore biodiversity, are high”; it is quite possible theoretically. But, once again, the sample (five localities) is too small to support this conclusion statistically. Pomacea is a very plastic species, able to adjust its life-cycle and physiology as an answer to various environmental challenges. An analysis of only five localities is not enough to cover all ecological diapason of this invader species.

I think that this MS, in the present form, cannot be accepted. However, if the authors can answer my objections concerning the design and basic assumption of their research, and prepare an improved version, based on an enlarged sample, it would be acceptable for publication.

My main objection is that the authors base their conclusions on a very limited sample, only five localities, including only two located within the native range of this snail (both situated in the same neighbourhood). In my opinion, the authors should determine whether there are no significant differences in trophic positions between several populations of the apple snails in its native area. Without this, all arguments about the difference between invasive and native populations lose all biological meaning. Insufficient sampling is a major cause of strong bias in ecological studies (even if their conclusions seem reliable and meaningful).

Another objection pertains to the selection of species evenness as a proxy for biotic homogenization. This selection must be substantiated in more details than it is made by the authors. First and foremost, even in natural communities, free from aliens’ impact, one can found many degrees of species evenness, from very high to very low. In other words, the use of this metric raises great doubts. The authors must prove that, in the sampled waterbodies, the presence of Pomacea was the only driver of species evenness.

Besides, there is a lot of minor remarks (see below).

Introduction: line 33 “Biodiversity [1]”. This reference looks very odd. For a biologist, “biodiversity” is also understandable and well-known, as for a physicist, the terms "force" or "atom". Is there a need to give a special reference to explain it?

Lines 33-34 “Biodiversity [1] encompasses the concepts of species diversity [2], taxonomic diversity [3], and community diversity [4]”. Why only these three facets of diversity are mentioned? What about functional diversity, for example? By the way, species diversity is a part of taxonomic diversity. Please, fix this mess.

Lines 46-47 “Invasive species often led to homogenization, a process which alters habitats by anthropogenic forces”. Homogenization of what? I’d recommend to add the word “biotic” before homogenization.

Lines 48-49 “This process, homogenization, generally increases the similarity of species”. Not “the similarity of species” but “similarity of species content” (of similarity of faunas, or floras).

The first two paragraphs of the Introduction (lines 33-53) contain a lot of quite trivial statements which are known to everyone in the field of invasion ecology, and, I am sure, the removing of part of these statements and corresponding references is very needed.

Lines 56-58 “Surprisingly, where P. canaliculata are naturally distributed [30], they represent major components of freshwater biodiversity, serve as ecosystem engineers, and contribute to nutrient turnover, while serving as prey for other species”. Why “surprisingly”? I cannot see any surprising in these facts. Many large species of freshwater snails have the same properties in their native areas.

Lines 67-69 “Comparisons of the trophic position of P. canaliculata between native and non-native habitats could reveal change in trophic position as a key factor influencing their adverse impacts as invasive species”. If this is a definition of the working hypothesis of this research, it must be done explicitly, and the goal(s) of the study clearly stated.

Material & Methods

Line 109 “Once collected, all specimens were inventoried, frozen, and thawed”. How much snails were collected from each site? What is the total number of studied individuals? Please, add this missing data to the M&Ms section.

Acknowledgments. Lines 452-453 “Last, I acknowledge funding from the Conchologist of America in support of this research in Hawaii”. The MS has three authors. Who of them is hidden under this “I”?

Author Response

Though this short text raises important problems and provides original and potentially interesting results, I do not feel that the conclusions presented by the authors are substantiated by the design of their study. In other words, the reported relationship (“a clear relationship between reduced biodiversity in habitats and the prevalence of adverse ecological impacts from invasive apple snails”) seems not to be properly substantiated by the facts and, thus, remains unconvincing. I can agree with this claim: “where species evenness values, and therefore biodiversity, are low <…> Pomacea canaliculata occupy trophic levels higher than corresponding snails collected in sites where species evenness values, and therefore biodiversity, are high”; it is quite possible theoretically. But, once again, the sample (five localities) is too small to support this conclusion statistically. Pomacea is a very plastic species, able to adjust its life-cycle and physiology as an answer to various environmental challenges. An analysis of only five localities is not enough to cover all ecological diapason of this invader species.

I think that this MS, in the present form, cannot be accepted. However, if the authors can answer my objections concerning the design and basic assumption of their research, and prepare an improved version, based on an enlarged sample, it would be acceptable for publication.

My main objection is that the authors base their conclusions on a very limited sample, only five localities,

If the collection of large numbers of plant, insect, arachnids, and molluscs, as well as other miscellaneous invertebrate species from 5 sites distributed over three continents (a vast geographical range), including the most isolated archipelago in the world, is considered a small sample. I would suggest that the number of invertebrates collected for these diversity insects and the comparable latitude of these non-native sites, in opposite hemispheres, clearly demonstrates the amount of time, planning, and thought that went into this experimental design.

including only two located within the native range of this snail (both situated in the same neighbourhood).

These two locations are very much different, one being a pristine lake containing thousands of apple snails another being an anthropogenically disturbed lake near the resort-laden coastline of Maldonado. These sites are separated by miles as indicated by the coordinates provided, so they ar not in the same,” neighbourhood”. A simple cut and paste of these coordinates would show the difference in locality but the difference in environmental continuity was evident. Perhaps the paper which was previously published as a result of this study, in this journal which we reference, would provide more detail into the variability between these two native sites if that was overlooked.

 In my opinion, the authors should determine whether there are no significant differences in trophic positions between several populations of the apple snails in its native area.

To be clear that would be of no real benefit to the study. The point was not if the native sites wear identical. Rather if the variability in biodiversity between these native sites, mainly caused by anthropogenic impacts, as well as differences in biodiversity in non-native habitats where apple snails exist correlated with changes in their trophic position. And to be clear, what you are suggesting was obviously done before funding was acquired to continue the study in China and Hawaii. One could pretty uch look at figure one and see these means would be different, an ANOVA would not be applicable as their variances are not equal but a non-parametric equivalent test of multiple means would be fine. This is what I did before spending four years on the last three collection sites.

Without this, all arguments about the difference between invasive and native populations lose all biological meaning.

I believe you mean ecological relevance, but as the previous comment stated. The native habitats were separated by miles and very much distinct from each other for the aforementioned reasons.

Insufficient sampling is a major cause of strong bias in ecological studies (even if their conclusions seem reliable and meaningful).

Again, I previously explained the differences between the native habitats being distinct and they were not in the same neighborhood.

The non-native habitats were selected on the basis of their similarity in longitude, the fact they were invaded by apple snails, and the fact they are on opposite sides of the planet.

I don’t see how you can question the reliability of my data, and whether or not my results are meaningful, if the statistics say otherwise. If your are honestly concerned with sample size you may want to note this study has a much higher sample size than any other study of apple snails using stable isotope data and measures of diversity in concert.

Another objection pertains to the selection of species evenness as a proxy for biotic homogenization.

From introduction,

This study employed the diversity index of species evenness [14-17], which defines the relative abundance of species, to measure the trophic position of the invasive apple snail Pomacea canaliculata in native and non-native habitats. The contributory relationships between biodiversity loss and altered ecosystem processes [18], function, and stability [19-21] became evident.

The selection of this index of diversity was based upon it mathematically being a measure of the similarity of species within a habitat (as the references provided state). This is literally a measure of diversity that on its lower end equates to homogenization.

 This selection must be substantiated in more details than it is made by the authors.

 First and foremost, even in natural communities, free from aliens’ impact (I suppose this means the adverse impact of biological invaders?), one can found many degrees of species evenness, from very high to very low.

In other words, the use of this metric raises great doubts. The authors must prove that, in the sampled waterbodies, the presence of Pomacea was the only driver of species evenness.

The authors never proposed that the presence of apple snails drive down diversity! To the contrary this study is concerned with assessing the trophic level of apple snails in habitats possessing various levels of biodiversity.

Besides, there is a lot of minor remarks (see below).

Introduction: line 33 “Biodiversity [1]”. This reference looks very odd. For a biologist, “biodiversity” is also understandable and well-known, as for a physicist, the terms "force" or "atom". Is there a need to give a special reference to explain it?

Sir or Madam, I see no reason for this comment. It is quite elitist.

I write in a manner where those who are not,” Biologists”, or even,” Physicist”, can understand the vernacular being employed. A sound understanding of the word is needed to comprehend the onus for this paper and as such is never an unsound idea.

Lines 33-34 “Biodiversity [1] encompasses the concepts of species diversity [2], taxonomic diversity [3], and community diversity [4]”. Why only these three facets of diversity are mentioned?

In the previous comment there was no need to explain biodiversity at all. It is curious that the number of facets of diversity here are limited to three. Perhaps, I did not want to descend into a diatribe and speak over the heads of layman readers.

 However, I will explain. As I am interested in the trophic ecology of invasive species and how that relates to diversity in native and non-native habitats my motivations would be as follows for the following types of biodiversity:

species diversity

The diversity of species is the foundation for sound ecological function. As species are the base unit for the formation of functional guild in ecosystems. These guilds perform various tasks to ensure the maintenance of systems and the regulated movement of energy, materials, and nutrients not just throughoutt out food webs but also throughout ecosystems both spatially and temporally.

taxonomic diversity

Taxonomic diversity allows one to consider the development and evolution of ecosystems alongside species. As we generally think of ecosystem on the scale of ecological timescales, 10,000’s to 100,000’s of years, it is important to remember that these ecosystems are unique amalgams of species. These species have no only evolved, but coevolved, with other sympatric species over evolutionary time scales. Though species may be introduced, the rate of species introduction into non-native habitats in higher now in the Anthropocene than at any other time in Earth’s known history. Confounding this is the rate of extinction in the Anthropocene, being as much as 100 times higher than any time in Earth’s known history.

community diversity

As we currently live in the Age of Man, also known as the Anthropocene, the loss of diversity globally is not only cause the loss of individual species, distinct and unique taxonomic groups, that may have evolved under rigid (stable) environmental conditions isolated from other land masses for example (e.g.: marsupials), but also the diversity we observe at the community level. This los f the diversity at the community level is termed homogenization.  

 What about functional diversity, for example?

 By the way, species diversity is a part of taxonomic diversity. Insinuating I am unaware of this, see above….

 Please, fix this mess.

SIR OR MADAM.

This comment is definitely out of line and has no constructive benefit. I do not know who this is behind the vail of anonymity you have been afforded. But, I am not your student, and from your tone I doubt I would want to be.

Please, in the future pretend to be civil.

Condescension is evident here and I will certainly make sure to bring this up with the journal.

“Besides, there is a lot of minor remarks (see below)”.

Lines 46-47 “Invasive species often led to homogenization, a process which alters habitats by anthropogenic forces”. Homogenization of what? I’d recommend to add the word “biotic” before homogenization.

I believe any Biologist would know that this term is synonymous with ecological studies.

Lines 48-49 “This process, homogenization, generally increases the similarity of species”. Not “the similarity of species” but “similarity of species content” (of similarity of faunas, or floras).

Don’t really understand why these petty and insult laden comments are necessary.

The first two paragraphs of the Introduction (lines 33-53) contain a lot of quite trivial statements which are known to everyone in the field of invasion ecology, and, I am sure, the removing of part of these statements and corresponding references is very needed. I disagree this is becoming quite derogatory. See prior comment

Lines 56-58 “Surprisingly, where P. canaliculata are naturally distributed [30], they represent major components of freshwater biodiversity, serve as ecosystem engineers, and contribute to nutrient turnover, while serving as prey for other species”. Why “surprisingly”? I cannot see any surprising in these facts. Many large species of freshwater snails have the same properties in their native areas. See comment prior

Lines 67-69 “Comparisons of the trophic position of P. canaliculata between native and non-native habitats could reveal change in trophic position as a key factor influencing their adverse impacts as invasive species”. If this is a definition of the working hypothesis of this research, it must be done explicitly, and the goal(s) of the study clearly stated.

Material & Methods

Line 109 “Once collected, all specimens were inventoried, frozen, and thawed”. How much snails were collected from each site? What is the total number of studied individuals? Please, add this missing data to the M&Ms section.

I believe you mean how many snails, everything you mention is in the supplemental data file. Had you read it.

Acknowledgments. Lines 452-453 “Last, I acknowledge funding from the Conchologist of America in support of this research in Hawaii”. The MS has three authors. Who of them is hidden under this “I”?

Rhetorical questions are for graduate students. This is not a constructive comment. Quite derogatory again.

I becomes we, there it is.

Reviewer 3 Report

The abstract should present the obtained results more clearly, without constantly resorting to statistical calculations.

In Material and working method, the overview of the order in which the observations and measurements were made and the role of each activity in the achievement of the objectives is required at the beginning, because the presentation is quite complicated. There is also the possibility of such an enlightening presentation at the beginning of each subchapter from Material and working method. The chapter lacks coherence.

L101-107 I don't think it matters how long it took to collect the evidence, nor how many people worked on it. The productivity of people at work is very different.

The trophic position of Pomacea canaliculata needs to be explained more clearly, because it generates a lot of confusion.

In the DISCUSSIONS section, in addition to the detailed analysis of the obtained data, the authors must compare their results with the results obtained by other authors. Researchers must explain how their findings align with or contradict existing research.

Author Response

The abstract should present the obtained results more clearly, without constantly resorting to statistical calculations.

Sir/ or Madam I have not resorted to statistical calculations. I have presented the pertinent results of well-known and easily understood statistical analyses in short for the reader in the abstract. While I do appreciate your time in reviewing the manuscript, I fail to see where my writing is ambiguous or unclear. Perhaps you could possibly specifically quote where you found my writing to be unclear? Thank you.

In Material and working method, the overview of the order in which the observations and measurements were made and the role of each activity in the achievement of the objectives is required at the beginning, because the presentation is quite complicated.

After reading this statement a few times, I believe you wish to say that the methods require a preamble explanation. I find this to be unnecessary as it would further complicate the methods section, as your comment seems to claim they are already too complicated to be understood. Curiously, this was the one review where that was the impression taken by the reader. But, thank you for your input.

There is also the possibility of such an enlightening presentation at the beginning of each subchapter from Material and working method. The chapter lacks coherence.

I do not agree with you that my writing is incoherent.

However, I do believe that it may be difficult for speakers of English as a second language to follow clearly. Nonetheless, I do not believe explaining my motivations for each method would make this section more,” coherent”, as it already is. Furthermore, I believe that the methods, as they are, explain very lucidly what was done; this is what methods should do.

L101-107 I don't think it matters how long it took to collect the evidence, nor how many people worked on it. The productivity of people at work is very different.

I believe the differences in productivity mentioned by you, may be attributable to the fact that each site had different densities of snails. As such, I thought it might be pertinent to explain, as your prior comment suggested, in detail differences in the time and number of personnel utilized in collections.        Again, methods should explain what was done. Without mentioning it it may be assumed the time and number of personnel would be identical between collection sites. I chose to make the implicit explicit here. Thank you.

The trophic position of Pomacea canaliculata needs to be explained more clearly, because it generates a lot of confusion.

I am not sure what you found confusing, as there was a formula provided as well as appendixes in the supplemental data for references to stable isotope values utilized in calculations. Nonetheless, again I am surprised it was not understandable to one of five reviewers. But, I think if you take the time to look again you may feel differently.

In the DISCUSSIONS section, in addition to the detailed analysis of the obtained data, the authors must compare their results with the results obtained by other authors. Researchers must explain how their findings align with or contradict existing research.

I will say this, a novel experiment is intended to create new knowledge and yes there are other studies that consider biodiversity and stable isotopes. Sadly, none have as large a sample size (please see appendixes for total sample size of metazoan species collected), nor are there any that use collection site spread over such as vast geographical area, nor are there any have calculated the trophic level of an invasive species; while attempting to control for environmental variability with stable isotope values from detrital material from respective collection sites.

Reviewer 4 Report

Review report of the manuscript “Assessing the impact of biodiversity (species evenness) on the trophic position of an invasive species (apple snails) in native and non-native habitats using stable isotopes”. 

Kevin E. Scriber, II1*, Christine A. M. France2 and Fatimah L.C. Jackson3

The authors provide actual and new results on the different impacts of the invasive snail Pomacea canaliculata trophic position on the biodiversity in native and non-native habitats. Data and analyses were presented appropriately. However, the manuscript contains some inaccuracies (listed below and marked in yellow in the manuscript), which corrections should improve the paper.

101 line – authors mentioned the collection of the “other sympatric animal species”. It is the first mention and some explanation about sympatric animals and their relation to snail species should be presented in the manuscript.

163-164 lines – not visible the numbers (marked in yellow)

176 line – the term “species catalog” is not precise, because only the part of collected organisms was identified to species level. As well, mentioned “detailed methods for the identification of species” were not found in the Supplementary sections.

230-233 lines – the title of Table 1 need to correct.

Supplementary files:

Section A.3 Table of List of all animals collected from Lake Sauce in Maldonado, Uruguay:

-authors mention that “All animal species cataloged and identified via COI barcoding in all collection sites during this study” but only part of the animals were identified as species.

-Why (on which basis) arachnids were separated from other macroinvertebrates and provided as the special group?

-The abbreviation sp. is used with the Latin name of the genus when the species is not known. With the common name of the class for example Arachnid, this abbreviation is not used. As well, an abbreviation is written in lowercase letters.

-Name “Odanata” shoud be corrected as Odonata.

-Use the uniform taxon for naming the groups of macroinvertebrates. Instead of shrimp include the class or order. The same comment is valid for the Apple snail.

-Use the uniform system to present the list of the collected invertebrates.

-”…..lowest taxonomical level (Genus and Family respectively)”, but the lowest taxonomic level is species.

-Repetition of the cells "Sample size (n)" or “Means of ID” is redundant.

-Compare and uniform both tables (List of all animals collected from Lake Sauce in Maldonado,

-Uruguay, and list of Lake Dario Maldonado, Uruguay), whereas the section of the "Field identification" is placed in the column "Means of ID" (lake Dario) meanwhile in the first table (lake Sauce) it has its own column "Field ID".

-Please, use the same terms, and taxon for the presentation of data: in the table List of all animals collected from Lake Sauce in Maldonado, Uruguay used Odonata species, meanwhile in the table (Lake Dario Maldonado, Uruguay ) already used Odonata larvae species...

-I missed the information on who and how identified the species (field ID). (table of the list species Lake Dario Maldonado, Uruguay)

-Does it possible to perform field identification with COI? (List of all animals collected in Kawainui Marsh)

-Table 3.14 and Appendix 2. are mentioned in the text (All species identifications from Kawainui Marsh are listed in Table 3.14 and the sequence data, by which these identifications were made, were listed in Appendix A.2.), but were not found.

Author Response

The authors provide actual and new results on the different impacts of the invasive snail Pomacea canaliculata trophic position on the biodiversity in native and non-native habitats. Data and analyses were presented appropriately. However, the manuscript contains some inaccuracies (listed below and marked in yellow in the manuscript), which corrections should improve the paper.

101 line – authors mentioned the collection of the “other sympatric animal species”. It is the first mention and some explanation about sympatric animals and their relation to snail species should be presented in the manuscript.

I agree, a brief explanation, or definition, of the term sympatric is warranted and I will acquiesce to your suggestion. Thank you.

163-164 lines – not visible the numbers (marked in yellow)

I believe I have resolved this issue. I did not find the highlighted numerals. If there is some mistake, I have overlooked, please make me aware. Thank you.

176 line – the term “species catalog” is not precise, because only the part of collected organisms was identified to species level. As well, mentioned “detailed methods for the identification of species” were not found in the Supplementary sections.

To your first point, the text has been changed to, “An animal collection catalogue “, and ,“animal collections catalogued” in the Methos and Supplemental Data sections respectively.

To your second point, the text in the Methods section was amended as follows:

“as well as the corresponding means of for the identification of animals collected and corresponding sequence data, where applicable, is available in Supplementary Materials sections A.3 and A.4 respectively”.

The Supplemental Data section was also amended as follows:

Section A.3: Animal collections catalogued and the corresponding means of for the identification of animals collected, and corresponding Cytochrome Oxidase Sub-unit I (COI ) sequence data, where applicable, from all collection sites during this study.

230-233 lines – the title of Table 1 need to correct.

Supplementary files:

Section A.3 Table of List of all animals collected from Lake Sauce in Maldonado, Uruguay:

-authors mention that “All animal species cataloged and identified via COI barcoding in all collection sites during this study” but only part of the animals were identified as species.

The prior comment and amendment render this comment mute; as the table mere states that it lists all the animals collected and it does so.

-Why (on which basis) were arachnids separated from other macroinvertebrates and provided as the special group?

The arachnids were separated on a taxonomical basis. The arachnids are clearly distinct from the hexapods collected morphologically. The main distinguishing characteristic between these groups being the synapomorphies that most easily identify them. That synopamorphy being the number of appendages they possess. As arachnids possess 8,” legs”, and insects possess,” 6”, they should logically be distinguished form each other in my opinion. Nonetheless, this why they were separated.

-The abbreviation sp. is used with the Latin name of the genus when the species is not known. With the common name of the class for example Arachnid, this abbreviation is not used. As well, an abbreviation is written in lowercase letters.

As, the identification of these organisms could not be classified to to genus, but these organisms were clearly different species based on morphological differences I used sp. To distinguish one from another.

As you sat it is not generally used in conjunction with class, with which I agree, what suggestion would you provide, as extensive attempts were made to identify these arachnids and insects to genus?

-Name “Odanata” shoud be corrected as Odonata.

A typo which is now corrected.

-Use the uniform taxon for naming the groups of macroinvertebrates. Instead of shrimp include the class or order. The same comment is valid for the Apple snail.

To your second point apple snails, is a general term, but after reading this introduction and methods it would be clear to the reader that the,” apple snail here”, is Pomacea canaliculata. In this table the common name was used to make it easy to distinguish the invasive species that the entire study centered on from other invertebrate species.

To your first point, yes shrimp is an easily understood common name. I was reluctant to use the Family Alpheidae, to which these shrimps belonged, for the reader to easily understand. I left the common name and provided the Family name, and subsequently the COI sequence data, to support that distinction.

I basically, did not want the reader to have to look up the family name of invertebrate species to have an idea of what they were. I did, however, provide said family name, or whatever taxonomical level of identification I could, to inform the reader more specifically.

-Use the uniform system to present the list of the collected invertebrates.

 See above comment. This may seem the best approach, but it would confuse any reader.

-”…..lowest taxonomical level (Genus and Family respectively)”, but the lowest taxonomic level is species.

This amendment has been made

lowest taxonomical level possible (ideally Family, Genus, species respectively)

-Repetition of the cells "Sample size (n)" or “Means of ID” is redundant.

I do not agree that this is,” redundant”. I believe the fact that this heading is present on each page prevents the reader from having to continuously con back to be assured of the information contained in each column.

-Compare and uniform both tables (List of all animals collected from Lake Sauce in Maldonado,

-Uruguay, and list of Lake Dario Maldonado, Uruguay), whereas the section of the "Field identification" is placed in the column "Means of ID" (lake Dario) meanwhile in the first table (lake Sauce) it has its own column "Field ID".

To be clear,” field Identification”, refers to the taxonomical ID assigned in the field by an Ichthyologist at the Cure Institute in Maldonado, Uruguay.

This was only done for Fish species….

These initial identifications were confirmed, or not, using COI.

This was explained in the Methods section I believe…..

-Please, use the same terms, and taxon for the presentation of data: in the table List of all animals collected from Lake Sauce in Maldonado, Uruguay used Odonata species, meanwhile in the table (Lake Dario Maldonado, Uruguay ) already used Odonata larvae species...

Sir or Madam, I appreciate your input but the reason why Odonata Species and Larvae were used is a matter of the different life stages these organisms were in as they were collected.

That is to say some Odonates were adults while others were aquatic larvae. This is all.

-I missed the information on who and how identified the species (field ID). (table of the list species Lake Dario Maldonado, Uruguay)

The Itchyologist who made these identifications was not interested in being part of the publication, as such I am not inclined to use their name.

-Does it possible to perform field identification with COI? (List of all animals collected in Kawainui Marsh)

Again, the field Identifications were made based on morphological characteristics, this time by me as these were snails and invertebrate species. The COI was used to confirm said identifications subsequently. You misunderstand.

-Table 3.14 and Appendix 2. are mentioned in the text (All species identifications from Kawainui Marsh are listed in Table 3.14 and the sequence data, by which these identifications were made, were listed in Appendix A.2.), but were

Where?

This should say Appendix A.4 and now does… You may find the formation you sought there. Thank you.

Reviewer 5 Report

The authors report on an interesting study of snail biodiversity using a stable isotope-based system. They reported that biodiversity, specifically species evenness, can have a significant impact on the trophic position of invasive species such as apple snails in both native and non-native habitats. Stable isotopes, which can provide insights into an organism's diet and trophic position, can be used as a powerful tool to assess these impacts.

The data reported by the authors show that in native habitats where biodiversity is generally higher, apple snails may encounter a more complex food web with a wider range of available food sources. This can lead to higher species evenness, meaning that different species within the community are present in relatively similar abundance. As a result, apple snails may have a more diverse diet and occupy different trophic positions, which can be reflected in their stable isotope signatures. For example, stable isotopes of carbon (δ13C) and nitrogen (δ15N) can be used to infer the primary producers (e.g., plants) at the base of the food web and the trophic level of the consumers (e.g., apple snails). In diverse habitats, apple snails may consume a wider range of food sources, resulting in a broader range of δ13C and δ15N values, indicating a higher trophic position.

On the other hand, in non-native habitats where biodiversity may be lower, apple snails may face reduced competition and encounter a simplified food web with limited food sources. This can result in lower species evenness and a narrower diet for apple snails, potentially leading to a more restricted trophic position. This can be reflected in their stable isotope signatures, with narrower ranges of δ13C and δ15N values, indicating a lower trophic position.

Additionally, invasive species like apple snails can sometimes undergo dietary shifts in non-native habitats due to changes in available resources and ecological pressures. This can further influence their stable isotope signatures and trophic position. For example, if apple snails in non-native habitats are primarily consuming a single food source, their stable isotope signatures may show a distinct shift compared to those in native habitats.

In summary, in my opinion, assessing the impact of biodiversity, specifically species evenness, on the trophic position of invasive species like apple snails using stable isotopes can provide valuable insights into their ecological role and interactions within native and non-native habitats. Higher species evenness in native habitats may result in more diverse diets and potentially higher trophic positions, while lower species evenness in non-native habitats may result in more restricted diets and potentially lower trophic positions. However, other factors such as dietary shifts and changes in resource availability should also be considered when interpreting stable isotope data for invasive species in different habitats.

The manuscript is clear and very important from an ecological point of view. After the authors correct the small typographical errors in the text, the manuscript can be published.

After the authors correct the small typographical errors in the text, the manuscript can be published.

Author Response

The authors report on an interesting study of snail biodiversity using a stable isotope-based system. They reported that biodiversity, specifically species evenness, can have a significant impact on the trophic position of invasive species such as apple snails in both native and non-native habitats. Stable isotopes, which can provide insights into an organism's diet and trophic position, can be used as a powerful tool to assess these impacts.

The data reported by the authors show that in native habitats where biodiversity is generally higher, apple snails may encounter a more complex food web with a wider range of available food sources. This can lead to higher species evenness, meaning that different species within the community are present in relatively similar abundance. As a result, apple snails may have a more diverse diet and occupy different trophic positions, which can be reflected in their stable isotope signatures. For example, stable isotopes of carbon (δ13C) and nitrogen (δ15N) can be used to infer the primary producers (e.g., plants) at the base of the food web and the trophic level of the consumers (e.g., apple snails). In diverse habitats, apple snails may consume a wider range of food sources, resulting in a broader range of δ13C and δ15N values, indicating a higher trophic position.

On the other hand, in non-native habitats where biodiversity may be lower, apple snails may face reduced competition and encounter a simplified food web with limited food sources. This can result in lower species evenness and a narrower diet for apple snails, potentially leading to a more restricted trophic position. This can be reflected in their stable isotope signatures, with narrower ranges of δ13C and δ15N values, indicating a lower trophic position.

Additionally, invasive species like apple snails can sometimes undergo dietary shifts in non-native habitats due to changes in available resources and ecological pressures. This can further influence their stable isotope signatures and trophic position. For example, if apple snails in non-native habitats are primarily consuming a single food source, their stable isotope signatures may show a distinct shift compared to those in native habitats.

In summary, in my opinion, assessing the impact of biodiversity, specifically species evenness, on the trophic position of invasive species like apple snails using stable isotopes can provide valuable insights into their ecological role and interactions within native and non-native habitats. Higher species evenness in native habitats may result in more diverse diets and potentially higher trophic positions, while lower species evenness in non-native habitats may result in more restricted diets and potentially lower trophic positions. However, other factors such as dietary shifts and changes in resource availability should also be considered when interpreting stable isotope data for invasive species in different habitats.

The manuscript is clear and very important from an ecological point of view. After the authors correct the small typographical errors in the text, the manuscript can be published.

I will address the error and hope to meet the appropriate standard thank you for reading the article.

Round 2

Reviewer 2 Report

Some of my criticisms were answered by the authors satisfactorily, some were not. I remain unconvinced that the 5-site study is properly designed to support the authors' conclusions. On the other hand, I can understand the authors' annoyance at my harsh criticism. However, it is normal for science when someone is not convinced by your results, isn't it? So I leave it to the discretion of the editor. I do not want to waste time on further debate with the authors. Maybe I and they see some things very differently. If the comments of other reviewers are more favourable to the content and conclusions of this study, let this MS be accepted. However, it is ridiculous to base conclusions about the invasive ecology of such a widespread alien species on only five sampling sites. However, in modern peer-reviewed journals on ecology, you can see something even more strange.

Author Response

Some of my criticisms were answered by the authors satisfactorily, some were not.

Some is plural. It seems that you have one criticism.

I remain unconvinced that the 5-site study is properly designed to support the authors' conclusions.

I see no specifics as to why you remain unconvinced based on my prior response.

On the other hand, I can understand the authors' annoyance at my harsh criticism. However, it is normal for science when someone is not convinced by your results, isn't it?

Irrelevant.

 So I leave it to the discretion of the editor.

You need a comma.

I do not want to waste time on further debate with the authors.

Your wasting time now.

Maybe I and they see some things very differently.

Irrelevant.

If the comments of other reviewers are more favourable to the content and conclusions of this study, let this MS be accepted.

“ So I leave it to the discretion of the editor.”

Irrelevant.

However, it is ridiculous to base conclusions about the invasive ecology of such a widespread alien species on only five sampling sites.

Irrelevant, “If the comments of other reviewers are more favorable to the content and conclusions of this study, let this MS be accepted.”   By the way, Ecology should always be capitalized.

However, in modern peer-reviewed journals on ecology, you can see something even more strange.

Irrelevant. Ecology.

Author Response

1) In the ABSTRACT it is also necessary to mention some concrete results from the paper.

Linear regression analysis established correlation between species evenness and apple snail trophic level (R2 = 0.8602); in line with a Pearson's product-moment correlation value (-0.83) and 95% confidence interval (-0.87, -0.77). Normal quartile plots indicated two normally distributed subsets of apple snail trophic level data: (1) a biodiverse subset containing the Uruguayan and Chinese lake sites and (2) the homogenized Hawaiian and Chinese creek sites. A precipice value for species evenness (separating biodiversity from homogenization), between (3.7) and (2.4), once descended to or surpassed separates statistically distinct, normal distributions of invasive apple snail trophic level data from diverse versus homogenized habitats.

The prior excerpt of text is taken directly from the abstract submitted. This except demonstrates the presence of result from this study; namely (1) a linear regression analysis, (2) Pearson’s Product correlation analysis, and (3) an interquartile plot ,of species evenness and/or calculated trophic level data from apple snails in all for 5 collection sites.

2)   STUDY AREA. I think the paper would be more complete if a brief climatic characterization of the studied area was made.

The crux of this manuscript is on the trophic ecology of invasive species, in native versus novel habitats, and the influence of biodiversity, as assessed by species evenness (as a measure of homogenization), on the trophic level occupied by said invasive species.

3) The DISCUSSIONS chapter lacks comparisons with other areas, with other similar research.

            To your point:

  • This is the first study I know of to compare the trophic ecology, specifically of invasive apple snails, over such a broad geographical range.

This range includes sites in:

  • The Southern hemisphere (Maldonado, Uruguay); within the native range of Pomacea canaliculata within the La Plata Basin at the base of the Pantanal.

  • The Hawaiian archipelago, the most isolated chain of islands on Earth, in the Western hemisphere; specifically on the island of Oahu in the culturally important site of Kawainui marsh.

  • And Xi Xi National Park in the Eastern hemisphere; specifically in the city of Hangzhou, Zhejiang, China. Here invasive apple snails have also invaded the famous West Lake Lotus Garden. However, this site was chosen as the plant diversity in West Lake was uniform and would therefore confound our interest in investigating the influence of biodiversity on the trophic level of this prolific invasive species, Pomacea canliculata.

4)   Is the suitability of some families/taxa/group of taxa as indicators of anthropogenic impact confirmed or not confirmed by other authors?

The taxonomic groups are not, in and of themselves, indicators of anthropogenic disturbance. Rather, these data just count data, as the taxonomical identifications of metazoans, including insects, fish, arachnids, oligochaetes, crustaceans, mollusks, etcetera, are count data used to calculate a value (species evenness) that indicates the heterogeneity of species within habitats.

It may seem odd that identifications were not to a uniform taxonomical level.

However, the field work, and subsequently, the lab work involved in this study have yielded taxonomic identifications that clearly differentiate metazoans collected as members of distinct taxonomic groups.

These identifications again, by themselves, are just count data. The actual measure of diversity is based on the relative abundance of the most abundant metazoan identified.

Perhaps the subsequent text was not meant for me.

- Why was the option of pollen analysis chosen to establish the indicators for the different degrees of anthropization and not directly of the plants that we can identify and quantify much more easily? This would possibly be justified if it were compared to previous periods of time when the anthropogenic impact in the Balearic Islands was much lower. That is, the spread of species from previous periods based on pollen, compared to the current situation. Fossil pollen is referred to in the paper, but comparisons of this pollen with the current one are not found in the results.

- The paper does not take into account the dynamics of vegetation and according to global warming in the last half century.

- Regarding the species that serve as agro-pastoral indicators, were there any differences compared to other previous periods or are they the ones that have always been there?

Round 3

Reviewer 3 Report

From my point of view, the paper has been sufficiently improved so that it can be published in this form.